# Physicochemical Profile of Antiandrogen Drug Bicalutamide: Solubility, Distribution, Permeability

**DOI:** 10.3390/pharmaceutics14030674

**Published:** 2022-03-18

**Authors:** Tatyana V. Volkova, Olga R. Simonova, German L. Perlovich

**Affiliations:** G.A. Krestov Institute of Solution Chemistry RAS, 153045 Ivanovo, Russia; vtv@isc-ras.ru (T.V.V.); ors@isc-ras.ru (O.R.S.)

**Keywords:** bicalutamide, solubility, Hansen parameter, distribution coefficient, thermodynamic approach

## Abstract

The pharmacologically relevant physicochemical properties of the antiandrogen drug bicalutamide (BCL) have been determined for the first time. Solubility in aqueous solution, 1-octanol, n-hexane, and ethanol was measured by the shake flask method in the temperature range of 293.15–313.15 K. The compound was shown to be poorly soluble in aqueous medium and n-hexane; at the same time, an essentially higher solubility in the alcohols was revealed. The following order of molar solubility was determined: ethanol > 1-octanol > water ≈ n-hexane. The solubility was correlated with the Van’t Hoff and Apelblat equations. Evaluation of the Hansen solubility parameters and the atomic group contribution approach of Hoftyzer and Van Krevelen demonstrated consistency with the experimental data and good potential adsorption of bicalutamide. The temperature dependences of the distribution coefficients in the 1-octanol/water and n-hexane/water two-phase systems were measured and discussed in the framework of the thermodynamic approach. The ∆logD parameter determined from the distribution experiment clearly demonstrated the preference of the lipophilic delivery pathways for the compound in the biological media. The overall thermodynamic analysis based on the solubility and distribution results of the present study and the sublimation characteristics published previously has been performed. To this end, the thermodynamic parameters of the dissolution, solvation, and transfer processes were calculated and discussed in view of the solute-solvent interactions. The permeation rate of BCL through the PermeaPad barrier was measured and compared with PAMPA permeability.

## 1. Introduction

Prostate cancer is one of the most common medical problems in the world, primarily due to its high incidence [1]. Up to 2012, bicalutamide and flutamide were the mainly used non-steroidal antiandrogen drugs. Bicalutamide (BCL) (Figure 1) (the brand name Casodex), among other anticancer drugs, is included in the World Health Organization’s list of essential medicines [2]. Its action is based on blocking the androgen receptor, the biological target of the androgen sex hormones testosterone and dihydrotestosterone [3]. Despite the development of several novel antiandrogen drugs, bicalutamide remains a therapeutic agent that is highly tradable on the drug market. However, the existing specific side effects complicate the choice of rational therapy [4] and propose ways to diminish the undesirable effects by selecting (design) the new promising structural analogues of the existent drug with a high therapeutic effect and low toxic action. In order to define the direction of the targeted synthesis, the overall description of the physicochemical properties (solubility, lipophilicity, diffusion rate) relevant for pharmaceutics and used to predict ADME (absorption, distribution, metabolism, excretion) characteristics of a drug should be disclosed. Moreover, the knowledge on the physicochemical profile of a drug compound may provide insights into potential issues expected in the clinic trials. That is why such parameters as solubility and distribution in pharmaceutically relevant solvents are considered to be among the properties in demand in the tests associated with pharmacokinetics and toxicological characteristics [5]. 

The solubility of a substance is determined by both the solid state (crystal lattice energy) and the interaction with the solvent (solvation) and plays an important role in drug transport and delivery [6]. In turn, the partition coefficient in the octanol/water system (log*P^oct/w^*) serves as a measure of the solvation energy [7]. This parameter can also be an important “indicator” of the partitioning of organic chemicals (including pharmaceuticals) between water and sediment [8] and is widely used to predict bioaccumulation [9].

It seems clear that the evaluation of the solubility and distribution coefficients within the scope of the relative Gibbs energy functions in the form of numerical values is not enough to disclose the interactions of the compounds with the solvents. To this end, the overall analysis of the mechanisms and driving forces of the processes in solutions from a thermodynamic point of view, using the respective enthalpy and entropic contributions to the Gibbs energy, is undoubtedly promising. Since the solvation processes cannot be evaluated directly from the experiments, quantitative information from the thermodynamic parameters in the solutions (solubility study) and in the solid state (sublimation experiments) can be successfully obtained. The solvation functions help to analyze the processes on an absolute energy scale. Moreover, these parameters comprise both specific and non-specific contributions, which can be split up using n-hexane—the solvent interacting with the drugs only non-specifically—as a standard. Additionally, the hypothetical transferring thermodynamic functions are very useful for disclosing the distribution between biological tissues. To this end, 1-octanol, aqueous solutions, and n-hexane are often applied as the solvents modeling the lipophilic medium of the biological cell membranes, blood plasma, and non-polar tissues (brain, for example), respectively.

In accordance with the extremely poor aqueous solubility recognized at approximately 3.7 μg·mL^−1^ [10] and 8.85 mg·L^−1^ [11] and good intestinal absorption, BCL is classified as a BCS (Biopharmaceutics Classification System) class II drug. Two BCL polymorphic modifications that differ from each other in stability and solubility in water have been proved [12]. Among them, form I was shown to be a more stable phase as compared to form II [13].

In the present study, we focused on the thermodynamic investigation of the dissolution and distribution processes of bicalutamide (BCL) form I in order to evaluate the transport properties of the drug, partitioning in the biological tissues, and diffusion through the biological membranes. The solubility and solvation thermodynamic parameters were determined in aqueous solution: 1-octanol, n-hexane, and ethanol (as a co-solvent and a preservative agent). Specific and non-specific impacts in the drug interaction with the solvents were disclosed using n-hexane, which interacts only by the non-specific forces. Furthermore, we analyzed and compared the transferring thermodynamic parameters derived from the solvation and distribution functions. To our knowledge, there is no information on BCL dissolution, solvation, and distribution thermodynamics in the literature. The permeability of BCL was measured through the lipophilic PermeaPad barrier.

The present study is a continuation of our investigations into the pharmacologically relevant physicochemical properties of drugs and drug-like compounds [14,15,16]. List of symbols:
*pK_a_*Ionization constant*S*_2_Molar solubility of solute (M)*X*_2_Mole fraction solubility of solute*M*_1_Molar mass of pure solvent (g·mol^−1^)*M*_2_Molar mass of solute (g·mol^−1^)*ρ*Density of pure solvents (g∙cm^−3^)*R*Universal gas constant (J⋅mol^−1^⋅K^−1^)*T*Temperature (K)*T_m_*Melting point (K)*p*Pressure (kPa)$\Delta H_{sol}^{0}$,$\Delta G_{sol}^{0}$,$\Delta S_{sol}^{0}$Standard dissolution enthalpy (kJ·mol^−1^), Gibbs free energy (kJ·mol^−1^), entropy (J⋅K^−1^⋅mol^−1^)$\Delta H_{solv}^{0}$,$\Delta G_{solv}^{0}$,$\Delta S_{solv}^{0}$Standard solvation enthalpy (kJ·mol^−1^), Gibbs free energy (kJ·mol^−1^), entropy (J⋅K^−1^⋅mol^−1^)$D_{C}^{oct/w}$,$D_{C}^{hex/w}$,$D_{C}^{oct/hex}$Apparent distribution coefficient in 1-octanol/water, n-hexane/water, 1-octanol/n-hexane system (molarity scale)$D_{X}^{oct/w}$,$D_{X}^{hex/w}$,$D_{X}^{oct/hex}$Apparent distribution coefficient in 1-octanol/water, n-hexane/water, 1-octanol/n-hexane system (mole fraction scale)$\Delta H_{tr}^{0}$, $\Delta G_{tr}^{0}$, $\Delta S_{tr}^{0}$Thermodynamic functions (enthalpy, Gibbs free energy, entropy) of transferring *J*Steady state flux through the membrane (µmol∙cm^−2^∙sec^−1^)*P_app_*Permeability coefficient (cm∙sec^−1^)u_r_(S)Relative standard uncertainties*u*(*T*)Standard uncertainty of temperature*u*(*p*)Standard uncertainty of pressure*Greek Letters*$\delta_{d}$, $\delta_{p}$, $\delta_{h}$Partial solubility parameters of Hansen$\Delta \delta_{t}$Total solubility parameter $\delta_{v}$Volume-dependent solubility parameter

## 2. Materials and Methods

### 2.1. Materials

Bicalutamide (BCL) (purity ≥ 98%) was purchased from Merck (KGaA, Darmstadt, Germany). 

1-octanol (purity ≥ 99%), n-hexane (purity ≥ 97%), and ethanol (purity 95.0%) were obtained from Sigma-Aldrich (St. Louis, MO, USA). 

Double distilled water (2.1 μS cm^−1^ electrical conductivity (PWT H198308 Pure Water Tester with Automatic Temperature Compensation, HANNA^®^ instruments, Karl Roth GmbH + Co 76,185 Karlsruhe, Tel. 0721/5 60 60. Resolution 0.1 μS cm^−1^)) was used for the experiments. 

All materials were used as received.

### 2.2. Methods

#### 2.2.1. Equilibrium Solubility Study: Dissolution and Solvation Thermodynamic Parameter Calculations

The standard shake-flask method [17] was applied to determine solubility. The saturated solutions of the studied compound were obtained in glass screw-capped vials with the respective solvent, which were vigorously stirred in an air thermostat. The volumes of water, 1-octanol, n-hexane, and ethanol equaled 7, 2, 7, and 2 mL, respectively. The time required for reaching equilibrium between the solute and the solvent was estimated with the help of the solubility kinetic dependences and was stated as 24 h for all solvents. After this time period, the saturated solutions stayed in the thermostat during the night, and then the non-dissolved substance was separated from the solution via centrifugation (Biofuge pico, Thermo Electron LED GmbH, Langenselbold, Germany) at the respective temperature for 20 min at 10,000 rpm. The concentrations of the compound in the saturated solutions were measured spectrophotometrically (Shimadzu 1800, Kyoto, Japan) at 5 temperatures, from 293.15 to 313.15 K (±0.1 K) using the calibration curves in each solvent at: *λ_max_* = 270 nm in water, 272 nm in 1-octanol and ethanol, and 267 nm in n-hexane. The experimental results were reported as an average of at least three replicated experiments with an accuracy of 2–4%.

The dissolution thermodynamic parameters on the standard state were determined using mole fraction unitary units. Recalculation of the molar solubility to the mole fraction was made as follows:(1)$$X_{2}=\frac{M_{1}S_{2}}{S_{2}(M_{1}-M_{2})+1000\rho_{}},$$
where *S*_2_ is the molar solubility of solute (M), *M*_1_ and *M*_2_ are the molar masses of solvent and solute, respectively, and *ρ* (g∙cm^−3^) is the density of pure solvents. Since the resulting solutions are very dilute, and their densities are very close to those of the pure solvents. The density values of the pure solvents were used in calculations [15] (Appendix A).

A linear function of the solubility temperature dependences within the chosen temperature interval expressed in mole fraction is described as:(2)$$\ln X_{2}=A+\frac{B}{T}$$

The linearity indicates that the variation of the heat capacity of the solutions with temperature is negligibly small. The temperature dependences of the experimental solubility were treated mathematically by the van’t Hoff equation, and the apparent solution enthalpies $\Delta H_{sol}^{0}$ were determined as follows:(3)$$\partial (\ln X_{2})/\partial T=\Delta H_{sol}^{0}/RT^{2}$$

The apparent free Gibbs energies of the solubility processes ($\Delta G_{sol}^{0}$) were calculated by the following equation:(4)$$\Delta G_{sol}^{0}=-RT(\ln X_{2})$$
where *X*_2_ and *R* are the drug molar fractions in the saturated solution and the universal gas constant, respectively. Next, the apparent solution entropies $\Delta S_{sol}^{0}$ were derived by the equation:(5)$$\Delta G_{sol}^{0}=\Delta H_{sol}^{0}-T\Delta S_{sol}^{o}$$

The solvation of one mole of the solute molecules in the solvent can be defined as the total change of the standard thermodynamic functions ($\Delta G_{solv}^{0}$, $\Delta H_{solv}^{0}$, $\Delta S_{solv}^{0}$) of the compound upon transferring it from the gaseous phase (ideal gas; single molecules without interaction) into the solvent. The transfer process can be presented as the difference between the dissolution and sublimation thermodynamic functions by:(6)$$\Delta Y_{solv}^{0}=\Delta Y_{sol}^{0}-\Delta Y_{sub}^{o}$$
where Δ*Y*^0^ is the standard change of any of the thermodynamic functions (Δ*G*^0^, Δ*H*^0^, Δ*S*^0^) of solvation ($\Delta Y_{solv}^{0}$), dissolution ($\Delta Y_{sol}^{0}$) or sublimation ($\Delta Y_{sub}^{o}$).

#### 2.2.2. Solubility Modeling with the Modified Apelblat Equation

In order to expand the quantitative description of the compound-solvent equilibrium at different temperatures, the temperature dependences of the experimental solubility were modeled using the modified Apelblat equation (in addition to the van’t Hoff one). To this end, the mole fraction BCL solubility (*X*_2_) in the studied solvents was calculated using Equation (7):(7)$$\ln X_{2}=A+\frac{B}{T}+C\ln T$$
where *A*, *B,* and *C* are the modeling empirical parameters: *A* and *B* reflect the specific features of the processes in solution coming from non-ideality, *C* includes the effect of temperature [18]; *T* is the absolute temperature expressed in K.

The relative average deviation (RAD) and the root-mean-square deviation (RMSD) were calculated in order to reveal the validity and accuracy of the models, as follows:(8)$$RAD=\frac{1}{N}\sum_{i=1}^{N}\left|\frac{X_{2}^{\exp}-X_{2}^{cal}}{X_{2}^{\exp}}\right|$$
(9)$$RMSD={\left|\frac{1}{N}\sum_{i=1}^{N}{\left(X_{2}^{\exp}-X_{2}^{cal}\right)}^{2}\right|}^{1/2}$$
where $X_{2}^{\exp}$ and $X_{2}^{cal}$ are the experimental and calculated solubility of BCL, respectively, expressed in mole fraction units, *N* means the number of the experimental points.

#### 2.2.3. Powder X-ray Diffraction (PXRD)

The powder XRD experiments were carried out on a D2 Phaser (Bragg-Brentano) diffractometer (Bruker AXS, Germany) with a copper X-ray source (λ_CuKα1_ = 1.5406 Å) and a high-resolution position-sensitive LYNXEYE XE-T detector under ambient conditions. The samples were placed into plate sample holders. The rotated speed was 15 rpm.

#### 2.2.4. Differential Scanning Calorimetry

The DSC thermograms of BCL pure and extracted from each solvent were obtained with the help of a Perkin-Elmer Pyris 1 DSC differential scanning calorimeter (Perkin-Elmer Analytical Instruments. Norwalk, CT, USA) with Pyris software for Windows NT in an atmosphere of dry helium of high purity (0.99996 mass fraction) (flowing 20 cm^3^∙min^−1^) using standard aluminum sample pans and a heating rate of 10 K∙min^−1^. The samples were weighed on an analytical balance (A&D GR-202, Japan) (uncertainty was 0.05 mg). The DSC was calibrated using a two-point calibration to measure the onset temperatures of indium and zinc standards. The onset of melting was used for calibration because it is almost independent of the scan rate. The melting temperatures for indium and zinc were 156.6 °C and 419.5 °C, respectively (determined by at least ten measurements). The obtained values exactly match recommendations [19]. The enthalpy scale was calibrated using the heat of the fusion of indium. The value obtained for the enthalpy of fusion was 28.69 J·g^−1^ (reference value is 28.66 J·g^−1^ [20]).

#### 2.2.5. Apparent Distribution Coefficients Determination and Transfer Thermodynamic Parameter Calculations

The well-known isothermal shake-flask method was applied for the determination of the apparent distribution coefficients in the 1-octanol/water ($D_{app}^{oct/w}$) and n-hexane/water ($D_{app}^{hex/w}$) systems. The mutually saturated solvents were prepared by mixing water and organic phases for 24 h and settling and separating from each other. The stock solutions of the drug were made from 1-octanol saturated with water for the 1-octanol/water distribution, and n-hexane saturated with water for the n-hexane/water system. An aliquot of the stock solution was placed in a glass vial together with an aliquot of the respective phase and mixed for 24 h at five temperatures, from 293.15 to 313.15 K (±0.1 K). The concentrations of the compounds were determined spectrophotometrically (Shimadzu 1800, Kyoto, Japan) in a UV region, with an accuracy of 2–4%. The reported experimental values represent the average of at least four replicate experiments.

The apparent distribution coefficients were calculated from the molar concentrations of the substance in the phases using the following equation:(10)$$D_{C}^{org/w}=\frac{C_{2}^{org/w}}{C_{2}^{w/org}}$$
where $D_{C}^{org/w}$ is the apparent distribution coefficient, $C_{2}^{org/w}$ and $C_{2}^{w/org}$ are the concentrations (M) of the compound after the distribution in the 1-octanol or n-hexane and water phases, respectively. The apparent distribution coefficients in the mole fraction scale ($D_{X}^{org/w}$) were calculated by the equation:(11)$$D_{X}^{org/w}=\frac{X_{2}^{org/w}}{X_{2}^{w/org}}$$
where $X_{2}^{org/w}$ and $X_{2}^{w/org}$ are the concentrations (mole fract.) of the compound after distribution. The standard Gibbs energy of the transfer ($\Delta G_{tr}^{0})$ from water to the organic phase was calculated as follows:(12)$$\Delta G_{tr}^{0}=-RT(\ln D_{X}^{org/w})$$

The transfer enthalpy ($\Delta H_{tr}^{0})$ was derived from the temperature dependences of the distribution coefficients by the van’t Hoff approach as:(13)$$\partial (\ln D_{X}^{org/w})/\partial T=\Delta H_{tr}^{0}/RT^{2}$$

The standard transfer entropy ($\Delta S_{tr}^{0}$) was determined by the equation:(14)$$\Delta G_{tr}^{0}=\Delta H_{tr}^{0}-T\Delta S_{tr}^{0}$$

#### 2.2.6. In Vitro Permeability Experiment

The permeability experiments were performed in a vertical-type Franz diffusion cell (PermeGear, Inc., Hellertown, PA, USA). The PermeaPad barrier (PHABIOC, Germany, www.permeapad.com, accessed on 20 May 2021) proposed by di Cagno et al. [21] was used as a lipophilic membrane between the two chambers of the Franz cell. The membrane effective surface area was 0.785 cm^2^. The experiment was carried out at 37.0 ± 0.1 °C. The detailed description and scheme of the reverse dialysis set-up employed in the present work can be found in our previous study [22]. The lower (donor) chamber of the Franz cell contained 7 mL of BCL aqueous donor solution (28.86 μg of BCL). The upper (receptor) chamber was filled with 1 mL of pure water. The donor solution was stirred with a magnetic stirrer bar (500 rpm). The samples from the receptor solution (0.5 mL) were withdrawn each 30 min and replaced with an equal amount of the respective fresh water. The experiment lasted 5 h. The lag time was 30 min. The concentrations of the sample solutions were measured spectrophotometrically (Spectramax 190; Molecular devices, Molecular Devices Corporation, California, CA, USA) in 96-well UV black plates (Costar) at *λ* = 270 nm. The amount of the permeated drug (dQ/A) was plotted versus time (t), taking into account the effective surface area of the membrane. A slope of the permeation plots produced flux (*J*):
(15)$$J\;=\;\frac{dQ}{A\;\times \;dt}$$

The slope (*J*), normalized by the concentration of BCL in the donor compartment (*C*_0_) gives the apparent permeability coefficient (*P_app_*): (16)$$P_{app}\;=\;\frac{J}{C_{0}}$$

The average value of *P_app_* from at least 3 experiments was taken into consideration.

## 3. Results

### 3.1. Solubility of BCL in Water and Organic Solvents

According to the literature [13], two crystalline forms (forms I and II) of bicalutamide (BCL) exist. In our study, we investigated form I, which was proven by the PXRD analysis (see Appendix A). The values of the BCL solubility measured in water, 1-octanol, n-hexane, and ethanol at different temperatures under atmospheric pressure are listed in Table 1. It should be emphasized that the mole fraction unitary units (mole fractions) are often better suited to the chemical interpretation of the solubility than the molarity (M) since the unitary quantities do not contain the mixing contribution, which is especially important in solvent effect studies [23]. To this end, both the molar scale (*S*_2_) and mole fraction (*X*_2_) solubility are introduced in Table 1. The solid phases after the solubility experiments were dried and subjected to PXRD and DSC analysis (Appendix A, respectively), which showed no solvate/hydrate formation or polymorphic transformations of BCL.

The *pK_a_* value of BCL was equal to 11.49 (calculated by ACD/LABS). This implies that the solubility in aqueous media is independent of pH in the biologically relevant region. The UV absorption spectra of the compound revealed the close similarity in all the solvents used: *λ_max_* = 270 nm in water, 272 nm in 1-octanol and ethanol, and 267 nm in n-hexane (Appendix A).

The BCL aqueous solubility measured in this work (Table 1) was in agreement with the values reported in our previous study [12] and the results of Mendyk et al. [10]. However, it is approximately 2-fold lower than was estimated by Patil et al. [11]. At the same time, Cockshott [24] reported the BCL solubility to be less than 5 mg∙L^−1^ which is close to our result. Probably, the discrepancies in the results of [11] from others can be attributed to the polymorphic form and physical state of BCL.

As follows from Table 1, very low BCL solubilities in water and n-hexane (1.47 × 10^−7^ and 1.07 × 10^−6^ mole fract., respectively, at 298.15 K) were estimated. A high polarity of water inhibits interactions with highly lipophilic BCL molecules, thus lowering the aqueous solubility. In turn, n-hexane (a non-polar hydrocarbon solvent) can interact with the drug molecules only by the London dispersion forces but not by the specific interactions. The solubility of the compound in the alcohols is high due to the strong dipole-dipole interactions of the alcohols with the lipophilic BCL molecule. The alcohol molecules consist of hydrophilic and hydrophobic fragments. The interaction with the hydroxyl groups of the alcohol induces a dipole moment in the BCL aromatic system, promoting the formation of hydrogen bonds. In addition, the interaction of the aromatic moiety of the substance with the nonpolar part of alcohol leads to solute—solvent dispersion forces. Among the alcohols, a 5.5-fold higher solubility value in ethanol (1.42 × 10^−3^ mole fract.) as compared to 1-octanol (2.57 × 10^−4^ mole fract.) was determined obviously due to a decrease in the polarity of the alcohol with the elongation of the alkyl chain [25] as a consequence of weakening the van der Waals intermolecular interactions with the solvent and reducing the hydrogen bonding ability of the dissolved compound. In all the solvents, the solubility increased with the temperature growth (Table 1), but this trend was the most pronounced in n-hexane.

### 3.2. Hansen Solubility Parameter for Solubility Prediction in Different Solvents

In the present study, the Hansen solubility parameters were used to evaluate the miscibility of BCL with the solvents. To this end, the group contribution methods of Hoftyzer-Van Krevelen and Fedors [26] were applied. These parameters assess the enthalpy contribution to the mixing energy, accounting for the cohesion energy as a measure of the intermolecular attraction in a respective liquid [27]. Notably, knowledge of the cohesive energies in active pharmaceutical ingredients allows the evaluation of their properties in respect to manufacturing processes, and their behavior inside the human body. Due to this, many applications of Hansen solubility parameters in both the pharmaceutical industry and drug administration were found [28]. The cohesion energies from the dispersion (*E_d_*), polar (*E_p_*), and hydrogen bonding (*E_h_*), as well as the molar volumes from the contributions of the fragments in the BCL molecule, are listed in Appendix A. The molar volumes and Hansen solubility parameters for BCL and the solvents are given in Table 2. The equations used for the calculations are placed in footnotes to the table.

The degree of the interaction between BCL and the solvents was evaluated using the $\delta_{t}$ parameter. The more closed the values of this parameter for the solute and the solvent are, the better miscibility (solubility) can be proposed. The $\Delta \delta_{t}$ parameter represents the difference between the $\delta_{t}$ parameters of the solute and the solvent [29]. The lower values of $\Delta \delta_{t}$ equal to 3.9 and 1.6 for ethanol and 1-octanol, respectively, indicate better solubility in these solvents as compared to water and n-hexane ($\Delta \delta_{t}$ = 25.2 and 7.7, respectively). In terms of the approach proposed in [30], the difference between the total solubility parameters <7 MPa^0.5^ (for ethanol and 1-octanol) means good miscibility with BCL, whereas $\Delta \delta_{t}$ > 10 MPa^0.5^ (for water) indicates that BCL is practically immiscible in aqueous solutions. Similar to BCL, a very low solubility for the structurally analogous nonsteroidal antiandrogen drugs—enzalutamide (3.92 × 10^−6^ M) and apalutamide (7.78 × 10^−6^ M)—at 298.15 K was estimated in our previous study [31]. Markedly, the values of the aqueous solubility of all three antiandrogen drugs meet well their respective (for aqueous solution) $\Delta \delta_{t}$ Hansen parameters: $\Delta \delta_{t}$ = 27.5 (enzalutamide) > $\Delta \delta_{t}$ = 25.2 (bicalutamide) ≈ $\Delta \delta_{t}$ = 24.8 (apalutamide). In spite of the fact that the solubility of BCL in n-hexane is very low, according to $\Delta \delta_{t}$ = 7.7 it falls in the range of average miscibility. Most probably, this discrepancy can be attributed to a different mode of interaction between the substance and n-hexane interacting only non-specifically as opposed to the other solvents used. In the next step, it was interesting to evaluate the impacts from different total specific parameters to $\Delta \delta_{t}$. As a result, the following order was estimated for BCL: $\delta_{d}$ > $\delta_{p}$ > $\delta_{h}$, evidencing the greatest potential of BCL from the interaction via the dispersion forces. The greatest difference between the $\delta_{h}$ terms of the total energy of BCL and water is apparently attributed to the high hydrophobicity of the BCL molecule responsible for its poor aqueous solubility. In their turn, from the very close values of the $\delta_{d}$ parameters for BCL and all the solvents used, an insignificant influence of the dispersion interaction on the solubility can be proposed. Moreover, just this fact can explain the poor solubility of the compound in n-hexane, where the dispersion interactions only are meaningful. It was interesting to analyze the impacts of different Hanson parameters for the alcohols (1-octanol and ethanol). A 1.6-fold greater (as compared to 1-octanol) hydrogen bonding potential ($\delta_{h}$) of ethanol is evident from Table 2, as a consequence of the lipophilicity of the 1-octanol molecule due to a lager (as compared to ethanol) alkyl chain. Moreover, ethanol demonstrates a 2.7-fold greater affinity to the polar interactions ($\delta_{p}$) with the compound caused by the permanent dipoles. As follows, a higher solubility of many compounds in ethanol was expected and proved for BCL in this study. At the same time, a discrepancy of the experimental solubility with the total solubility parameter ($\Delta \delta_{t}$) is observed for 1-octanol (*X*_2_ = 2.57 × 10^−4^ mole fract., $\Delta \delta_{t}$ = 1.6) and ethanol (*X*_2_ = 1.42 × 10^−3^ mole fract., $\Delta \delta_{t}$ = 3.9). It can be suggested that not only the molecular structures of the compound and the solvents but also the mutual influence of the other factors, such as an additional impact of the solute-solute and solvent-solvent interactions on the solvation phenomenon, as well as a geometric configuration of the species in the solutions, etc., can be also meaningful.

In addition, in order to estimate the affinity between BCL and the solvents, a Relative Energy Difference (RED) was calculated as:(17)$$RED=\frac{R_{a}}{R_{0}}$$
where *R_a_* and *R*_0_ are the solubility “distance” parameter between BCL and solvents and the interaction radius of BCL, respectively [32]. *R_a_* and *R*_0_ were determined by the equations [33]:(18)$${\left(R_{a}\right)}^{2}=4{\left(\delta_{d2}-\delta_{d1}\right)}^{2}+{\left(\delta_{p2}-\delta_{p1}\right)}^{2}+{\left(\delta_{h2}-\delta_{h1}\right)}^{2}$$
(19)$$R_{0}={\left(\delta_{d1}{}^{2}+\delta_{p1}{}^{2}\right)}^{0.5}-\delta_{h1}$$
where indexes 1 and 2 refer to BCL and solvent, respectively. *R_a_*_,_ *R*_0_ and *RED* are given in Table 2. *RED* values below 1 show good BCL affinity to 1-octanol and n-hexane. As opposed, low BCL affinity to n-hexane and, especially, water was proven (*RED* > 1).

It was emphasized that the Hansen parameters, reflecting the dispersion ($\delta_{d}$) and polar ($\delta_{p}$) forces, have a similar nature, which is different from the hydrogen bonding ($\delta_{h}$). A very useful tool often used to discriminate the hydrogen bonding from the dispersion and polar forces is a volume-dependent $\delta_{\upsilon }={\left(\delta_{d}^{2}+\delta_{p}^{2}\right)}^{1/2}$ parameter (Table 2) and a Bagley diagram proposed by Bagley et al. [34] representing a dependence of *δ_v_* against *δ_h_*. The Bagley diagram for BCL is illustrated in Figure 2. The Bagley diagram visually demonstrates the regularities and the discrepancies disclosed from the Δ*δ_t_*—parameters. Importantly, for BCL *δ_v_* = 20.8 and *δ_h_* = 9.0 are close to the region characteristic for the optimal absorption after oral administration estimated by Breitkreutz [35] (*δ_v_* = 20.3 MPa^0.5^ and *δ_h_* = 11.3 MPa^0.5^), indicating a close proximity to the region where the optimal absorption characteristics in the case of oral administration are located.

### 3.3. Modeling of Solubility Data by Van’t Hoff and Modified Apelblat Equations

The relationship between the experimental solubility expressed in mole fractions and the temperature was estimated by fitting with the van’t Hoff and the modified Apelblat equations (Equations (3) and (7)). The experimental solubility values, calculated ones, and relative deviations are tabulated (Appendix A), and the parameters of the modified Apelblat and van’t Hoff equations are listed in Appendix A. As follows from Appendix A, the reliable correlations between the experimental and calculated (by modeling with the van’t Hoff and Apelblat equations) values of BCL solubility were derived. Both models are appropriate for modeling the BCL solubility in all the investigated solvents according to the total average RD, RAD, and RMSD parameters.

### 3.4. Thermodynamics of Solubility, Solvation, and Transfer Processes

The thermodynamic functions of the dissolution at 298.15 K calculated from the van’t Hoff plots (Figure 3) are presented in Table 3.

From the thermodynamic point of view, the driving force of the dissolution ($\Delta G_{sol}^{}$) is a combination of both the enthalpy and entropy contributions [36]. As shown in Table 3, the values of the solution enthalpies ($\Delta H_{sol}^{0}$) are positive in all the solvents (endothermic process) as a consequence of a larger crystal lattice energy of BCL as compared to the solvation energy. The enthalpy factor (by the absolute value) exceeds essentially the entropy one in all the studied solvents, showing the enthalpy determined dissolution process. The positive entropy values in the organic solvents impact favorable dissolution (entropy-driven process) [36]. At the same time, a rather high value of $\Delta H_{sol}^{0}$ in n-hexane levels the entropy effect and leads to low solubility in this solvent. The highest BCL solubility in ethanol is determined by the optimal (among the other solvents) balance between the enthalpy and entropy contributions to the driving force of the dissolution process. As opposed, the dissolution entropy in water is negative most likely due to the hydrophobic effects (an essential ordering of the water molecules). These facts, as well as the high value of the enthalpy term, hamper the dissolution process and lead to extremely poor BCL aqueous solubility. In the next step, the solvation thermodynamic functions (Table 3) were calculated by Equation (6) taking into account the sublimation thermodynamic functions from our previous study [12] (Appendix A) in order to access the interactions of the drug with the solvents on the absolute energy scale. According to the absolute values of the $\Delta H_{solv}^{0}$ parameter, the solvents ranged as follows: 1-octanol > ethanol > water > n-hexane. Evidently, this order differs from the solubility regularity. At the same time, the order of the solvents with respect to the driving force of the solvation process at a standard temperature of 298.15 K ($\Delta G_{solv}^{0}$) corresponds to the solubility trend: ethanol > 1-octanol > n-hexane > water. This fact testifies on behalf of some impact of the entropy factor in $\Delta G_{solv}^{0}$. In order to quantitatively disclose this impact, the contributions of the enthalpy and entropic solvation terms were evaluated using the $\zeta_{H}{}_{solv}^{}$ and $\zeta_{TS}{}_{solv}^{}$ parameters (Table 3). The analysis of these values demonstrates the dominant role of the solvation enthalpy to $\Delta G_{solv}^{0}$ ($\zeta_{H}{}_{solv}^{}$ > $\zeta_{TS}{}_{solv}^{}$). Meanwhile, in water, the contributions from $\Delta H_{solv}^{0}$ and $T\Delta S_{solv}^{0}$ are closer than those in the other solvents in which $\zeta_{H}{}_{solv}^{}$ > $\zeta_{TS}{}_{solv}^{}$ up to 2-fold (in ethanol), which once more proves an essential role of the hydrophobic effects in the BCL solvation in aqueous solution. In addition, the dissolution and solvation thermodynamic parameters were calculated at all studied temperature points (Appendix A). The temperature effect on the change of the dissolution and solvation of Gibbs energy (Δ*G*) and entropy (*T*Δ*S*) was shown to be the most pronounced for the solubility in n-hexane. In water and 1-octanol, the impact of the temperature was shown to be negligible. Interestingly, the solvation entropy term $\zeta_{TS}{}_{solv}^{}$ decreases with the temperature growth (demonstrating the ordering of the systems at higher temperatures) in all solvents except water.

The impact of the specific interactions on the solvation process was estimated by analyzing the thermodynamic functions of the hypothetical transfer from n-hexane—an inert solvent interacting with the drugs only non-specifically—to water, 1-octanol, and ethanol. Besides this, the hypothetic transfer from water to 1-octanol is significant as a model of the drug penetrating through the lipophilic biological membranes. The transferring thermodynamic parameters listed in Table 4 were calculated as a difference (taking into account the sign of the thermodynamic function) between the respective functions in a solvent (1-octanol/water/ethanol) and in n-hexane and between 1-octanol and water.

As follows from the results (Table 4), according to the sign of the driving force (Δ*G_tr_*) and ($\zeta_{Htr}$ and $\zeta_{TS}{}_{tr}^{}$) parameters, the transfer process is favorable and enthalpy determined in all systems except the (n-hexane → water) one. At the same time, only the (water → 1-octanol) transfer is driven by the entropy term (*T*Δ*S_tr_* = 3.3 kJ⋅mol^−1^), most likely as a result of the dominant role of the hydrophobic effect and disordering of the 1-octanol phase. Moreover, according to negative Δ*H_tr_*, the interactions of the BCL molecules with the solvate shell in 1-octanol are stronger as compared to water, and there is a high probability of the re-solvation of the drug molecules during the (water → 1-octanol) transfer.

The relationship between the specific and non-specific solvation thermodynamic contributions was disclosed using the $\varepsilon_{H}$ and $\varepsilon_{S}$ parameters calculated as follows:(20)$$\varepsilon_{H}=(\Delta H_{spec}/\Delta H_{non-spec})\cdot 100\%\;\mathrm{where}\;\Delta H_{spec}=\Delta H_{tr}(n-hexane\to solvent)\mathrm{and}\;\;\Delta H_{non-spec}=\Delta H_{solv}^{}(n-hexane)$$
(21)$$\varepsilon_{S}=(\Delta S_{spec}/\Delta S_{non-spec})\cdot 100\%\;\mathrm{where}\;\Delta S_{spec}=\Delta S_{tr}(n-hexane\to solvent)\mathrm{and}\;\;\Delta S_{non-spec}=\Delta S_{solv}^{}(n-hexane)$$

The $\varepsilon_{H}$ parameter is the maximal for the solvation of BCL in ethanol, whereas the minimal value is characteristic for the aqueous solution. It seems reasonable that the specific interactions in ethanol are more pronounced, and this is in a consequence with the highest solubility in this solvent. For the solvation in 1-octanol, the impacts of the specific interactions in both $\varepsilon_{H}$ and $\varepsilon_{S}$ parameters have very close values. At the same time, the main contribution of the specific forces to the entropy parameter (at approximately 2-fold greater) as compared to the hydration enthalpy in water was estimated, obviously due to the disordering of the water molecules in the solvation shell of BCL. The obtained information seems to be important since it allows for deeper insight into the nature of the intermolecular interactions during the solvation process.

### 3.5. Apparent Distribution Coefficients in 1-Octanol/Water and n-Hexane/Water Systems

The distribution coefficients in the systems of the mutually saturated biologically relevant model solvents (water, 1-octanol, n-hexane) are useful for disclosing the nature of the processes on the border between the biological tissues. The distribution coefficient in the 1-octanol/water system ($\log D^{oct/w}$) corresponds to a very important parameter named lipophilicity and serves as a characteristic of the passage of the drug molecules from the bloodstream to the lipophilic layers of the intestinal membranes [37]. From the thermodynamic point of view, it can be considered as a measure of the solvation energy, which along with the crystal lattice energy is the determinative factor for the solubility of drug. Another parameter usually applied in order to disclose the overall distribution behavior of the drug compound is the distribution coefficient in the n-hexane/water (or cyclohexane/water) system ($\log D^{hex/w}$). This parameter allows, firstly, to evaluate the penetrating of the substance into the non-polar biological tissues (e.g., brain) and, secondly, to assess the hydrogen-bonding ability of a solute and the specific interactions with the solvents. The second issue can be disclosed via the Δlog*D* term calculated as follows [38]:(22)$$\Delta \log D=\log D_{C}^{oct/w}-\log D_{C}^{hex/w}{}^{}$$
where $\log D_{C}^{oct/w}$ and $\log D_{C}^{hex/w}{}^{}$ are the distribution coefficients in 1-octanol/water and n-hexane/water systems, respectively, calculated from the molar (M) concentrations of the solute. In this study, the apparent distribution coefficients of BCL in the 1-octanol/water and n-hexane/water systems were determined at different temperatures. The distribution coefficients calculated by Equation (10) are presented in Table 5 for T = 298.15 K.

The BCL distribution coefficient in the 1-octanol/water system at 298.15 K log$D_{C}^{oct/w}$ = 2.82 (Table 5) correlates with the value of log$D_{}^{oct/w}$ = 2.54 published in [8] and the calculated value (log*D^oct/w^* = 2.71) from ChemAxon. As the results in Table 5 show, the distribution equilibrium in both 1-octanol/water and n-hexane/water systems is shifted to the organic phase. At the same time, $D_{C}^{oct/w}$ is 352-fold higher than $D_{C}^{hex/w}$ due to the stronger interaction of the lipophilic BCL molecules with 1-octanol as compared to n-hexane. In addition, according to Kerns and Di’s classification [5], log$D_{C}^{oct/w}$ = 2.82 belongs to the optimal potential range of the compound bioavailability (log*D* = 1 ÷ 3), and the value of ∆log*D* equal to 2.55 allows the affinity of the investigated substance to the lipophilic regions rather than the non-polar biological tissues to be proposed. For the sake of comparison, the distribution coefficients for the structurally parent compounds enzalutamide and appalutamide (log$D_{C}^{oct/w}$ = 4.02 and 3.90, respectively) [31] go beyond the range of ideal lipophilicity. The experimental distribution coefficients ($D_{C}^{org/w}$) have been compared with the values calculated as the ratios of the solubilities in 1-octanol or n-hexane and water ($D_{C/calc}^{org/w}$). The calculated coefficients at 298.15 K are listed in Table 5. As follows, the experimental values exceeded the calculated ones: 3.3- and 1.9-fold in the 1-octanol/water and n-hexane/water systems, respectively. This fact clearly demonstrates the impact of the mutual saturation of the solvents on the distribution results. Nevertheless, according to the calculated log$D_{C}^{oct/w}$ = 2.30, the investigated compound also belongs to the “ideal” range of lipophilicity. Interestingly, only a slight difference (1.1-fold log units) between the ∆log*D* parameters derived from the experimental and calculated distribution coefficients exists.

### 3.6. Transfer Thermodynamics

It is possible for compounds to have similar distribution coefficients but rather different mechanisms of the partitioning, which can be disclosed from the respective thermodynamic parameters (enthalpies and entropies). These parameters can be obtained from the temperature dependences of the distribution coefficients. The temperature dependences of the BCL experimental distribution coefficients in the 1-octanol/water ($D_{X}^{oct/w}$), n-hexane/water ($D_{X}^{hex/w}$), and 1-octanol/n-hexane ($D_{X}^{oct/hex}$) systems expressed in the mole fraction unitary units (Equation (11)) are given in Figure 4 in the natural logarithmic scale. Figure 4 shows the decrease of $D_{X}^{oct/w}$ and $D_{X}^{oct/hex}$, as well as the increase of $D_{X}^{hex/w}$ with the temperature growth. The transfer thermodynamic parameters determined by Equations (12)–(14) are listed in Table 6. Notably, these parameters characterize a “real” transfer process (from the distribution experiments) as opposed to the hypothetical transfer disclosed from the solvation thermodynamic functions (see Section 3.3, Table 4).

Table 1 octanol or n-hexane is a spontaneous process, as indicated by negative $\Delta G_{tr}^{0}$. Expectedly, the free energy of the transfer into 1-octanol is greater than into n-hexane due to the higher solubility and stronger interactions of BCL with 1-octanol. However, the enthalpy and entropy parts of $\Delta G_{tr}^{0}$ are greatly different in these systems (Table 6). 1-Octanol has an exothermic heat of transfer (negative enthalpy) due to the hydrogen-bond stabilization of the transferred substance, not found in n-hexane [39]. Although the transfer enthalpy term in the (water → n-hexane) transition is positive (endothermic process), the entropy increase is somewhat greater, enough to offset the enthalpy and to result in an entropy-driven process determined by the hydrophobic effect.

The comparative analysis revealed that the hypothetical transfer functions (derived from the solvation parameters; see Section 3.3 Table 4) agree with the parameters derived from the distribution experiments (“real” transfer) (Table 6). In order to illustrate the regularities following from the comparison of the enthalpy and entropy contributions to the transfer Gibbs energy, the diagram approach was applied (Figure 5). A detailed description of the regions on the diagram has been reported by us before [14].

It is evident from the diagram that the transferring thermodynamic functions of the hypothetical and “real” processes are in good agreement. Very close values of $\Delta G_{tr}$ for the hypothetical and “real” processes (−18.5 and −21.5 kJ⋅mol^−1^, respectively) were obtained for the (water → 1-octanol) transfer which is the enthalpy-determined process (sector C). The slight difference of the $\Delta G_{tr}$ values in this system comes from the entropy terms $T\Delta S_{tr}^{0}$ (“real”) ≥ $T\Delta S_{tr}^{0}$ (hypothetical). A low value of the entropic term ($T\Delta S_{tr}^{0}$ = 3.3 ÷ 6.5 kJ⋅mol^−1^) means that the transfer in this system practically does not change the ordering of the solvent molecules in the solvation shell. The lower driving force of the transfer from water to n-hexane (as compared to 1-octanol) equals to $\Delta G_{tr}$ = −4.8 (hypothetical) and −6.5 (“real”) kJ⋅mol^−1^, and the process is determined and driven by the entropy (sector A). The small $\Delta G_{tr}$ value testifies to the insignificant driving force for the transition between the aqueous media and the non-polar biological tissues (for example, the brain). In turn, the driving force for the (water → 1-octanol) transfer is approximately 3.9 ÷ 3.3-fold greater, allowing a successful transition of the drug from the aqueous medium of the blood plasma through the lipophilic layers of the biological membrane and, as a consequence, good intestinal absorption.

As mentioned above, the ∆log*D* parameter is a very useful “indicator” of the hydrogen-bonding ability and the specific interactions of a solute with the solvents and can characterize the transfer of Gibbs energy from n-hexane to 1-octanol. As follows, additional information on the thermodynamic functions of this parameter (enthalpies and entropies) was obtained from the temperature dependences in order to disclose the transfer (n-hexane → 1-octanol) thermodynamic mechanisms. To this end, the ∆ln$D_{X}^{}$ values (expressed in mole fraction units) characterizing the transfer (n-hexane → 1-octanol) were calculated as ∆ln *D_X_* = ln$D_{X}^{oct/w}$ − ln$D_{X}^{hex/w}$ and the respective thermodynamic functions were determined using Equations (12)–(14). The resulting temperature dependences and the contributions of the enthalpy and entropy terms to the free Gibbs energy of the hypothetical and “real” transfers are shown in Figure 5 and Table 6. Figure 5 demonstrates good agreement between the hypothetical and “real” transfer thermodynamic parameters. At the same time, the thermodynamic mechanism of the (n-hexane → 1-octanol) transfer differs from the (water → organic solvents) transitions. In particular, the (n-hexane → 1-octanol) transfer process is enthalpy determined (|$\Delta H_{tr}^{0}$| > |$T\Delta S_{tr}^{0}$|) similar to the (water → 1-octanol) one. Nevertheless, $T\Delta S_{tr}^{0}$ < 0 testifies to the ordering of the system as opposed to water → 1-octanol and water → n-hexane ($T\Delta S_{tr}^{0}$ > 0). It can be concluded that the mechanism of the (n-hexane → 1-octanol) transfer is absolutely different from the (water → n-hexane) and partially agrees with the (water → 1-octanol) transition. Importantly, the regularities estimated from the transferring diagram analysis match the results obtained using the Hansen parameters ($\Delta \delta_{t}$^water^ > $\Delta \delta_{t}$^n-hexane^ > $\Delta \delta_{t}$^1-octanol^).

### 3.7. Permeation of BCL through the PermeaPad Barrier

In order to complete the physicochemical profile of BCL, in the last step of our study, we measured the permeability of the compound through the PermeaPad barrier using the vertical-type Franz diffusion cell. The kinetic dependence of the cumulative amount of BCL permeated through the PermeaPad barrier is illustrated in Appendix A. The concentration of BCL in the stock (donor) solution was very close to its aqueous solubility at 37 °C (*C* = 9.58 × 10^−6^ M), the steady-state flux was determined from the slope of the kinetic dependence of the cumulative amount of the permeated drug (Equation (15)) (*J* = 1.59 × 10^−7^ µM∙cm^−2^∙s^−1^). The permeability coefficient was calculated by Equation (16) and was shown to be *P_app_* = (1.66 ± 0.14) × 10^−5^ cm∙s^−1^ (log*P_app_* = −4.78). It was interesting to compare the BCL permeability through the PermeaPad barrier from our study with the respective information on the permeability from other well-known models. According to the study of [40], the permeability measured for bicalutamide with the help of the parallel artificial membrane permeability assay (PAMPA) *P_app_* = 7.76 × 10^−6^ cm∙s^−1^ (log*P_app_* = −5.11) is at approximately 2.1-fold lower than our PermeaPad result, most probably due to the differences in nature of the lipids in the content of the artificial membranes and the experimental set-up.

We assume that the information on the BCL permeability coefficient could be useful for evaluating the permeability in the case of the preparation of water-soluble formulations based on BCL using different approaches, for example, co-crystals with suitable co-formers and solid dispersions with cyclodextrins and biopolymers (the objects of our future studies).

## 4. Conclusions

New experimental results were obtained in order to disclose the physicochemical profile of antiandrogen drug bicalutamide (BCL) crystalline form I. The solubility in aqueous solution, 1-octanol, n-hexane, and ethanol was determined in the temperature range of 293.15−313.15 K. Very low BCL solubility in water and n-hexane (1.47 × 10^−7^ and 1.07 × 10^−6^ mole fract., respectively) was estimated at 298.15 K. The solubility in alcohols was shown to be significantly higher: 1.42 × 10^−3^ mole fract. and 2.57 × 10^−4^ mole fract. in ethanol and 1-octanol, respectively. In all the solvents, the solubility increased with the temperature growth, but this trend was the most pronounced in n-hexane. The experimental solubility was successfully approximated by two thermodynamic models using the van’t Hoff and modified Apelblat equations. The thermodynamic parameters of the dissolution, solvation, and hypothetical transfer from n-hexane to the other solvents and from water to 1-octanol were calculated. The positive entropy values in the organic solvents impact favorable dissolution (entropy-driven process). In contrast, the dissolution entropy in water is negative due to an essential ordering of the water molecules (hydrophobic effects). Just this fact, as well as a high value of the enthalpy term, was shown to be the main reason for hampering the dissolution process and poor BCL aqueous solubility.

The temperature dependences of the apparent distribution coefficients in the 1-octanol/water and n-hexane/water two-phase systems were measured. The BCL distribution coefficient in the 1-octanol/water system was shown to belong to the optimal potential range of the compound bioavailability, and the value of ∆log*D* allowed us to propose the affinity of the investigated substance to the lipophilic regions rather than the non-polar biological tissues.

The agreement between the thermodynamic parameters of the hypothetical (from solvation) and “real” (from distribution) water → 1-octanol, water → n-hexane, and n-hexane → 1-octanol transfers was revealed with the help of the diagram approach. The driving forces of the transfer processes were disclosed. According to the driving forces, the results of the transferring diagram analysis match the regularities estimated with the help of the Hansen solubility parameters ($\Delta \delta_{t}$^water^ > $\Delta \delta_{t}$^n-hexane^ > $\Delta \delta_{t}$^1-octanol^). It was concluded that the mechanism of the (n-hexane → 1-octanol) transfer is absolutely different from the (water → n-hexane) transfer and partially agrees with the (water → 1-octanol) transition.

The BCL permeability coefficient determined through the PermeaPad barrier was shown to be 2.1-fold higher compared to PAMPA reported in the literature.

## Figures and Tables

**Figure 1 pharmaceutics-14-00674-f001:**
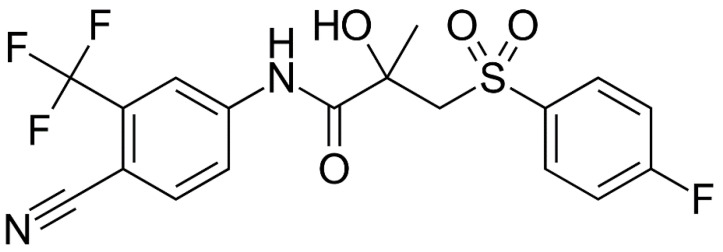
The structure of bicalutamide (BCL).

**Figure 2 pharmaceutics-14-00674-f002:**
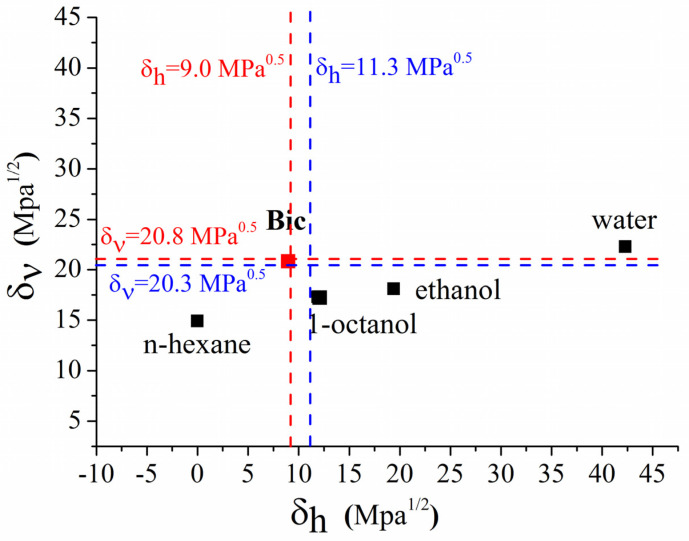
Bagley diagram for BCL. Red and blue dashed lines indicate the Breitkreutz [34] parameters for BCL and those parameters characteristic for the optimal absorption.

**Figure 3 pharmaceutics-14-00674-f003:**
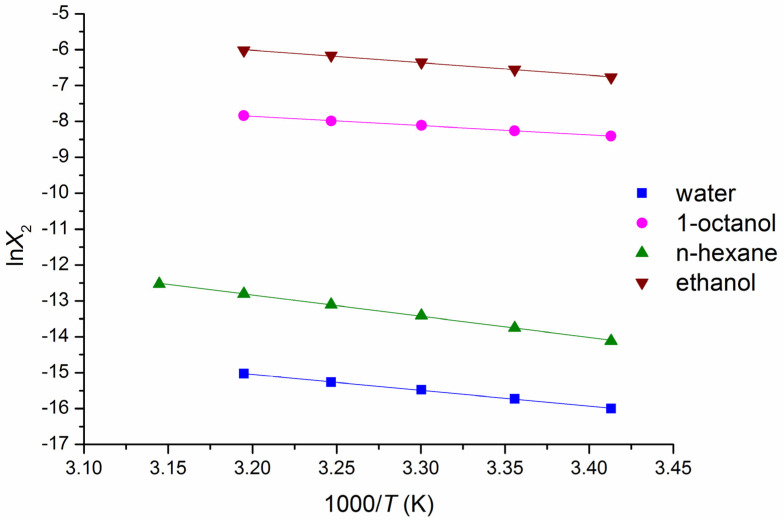
Temperature dependences of the BCL solubility in the investigated solvents (mole fraction scale).

**Figure 4 pharmaceutics-14-00674-f004:**
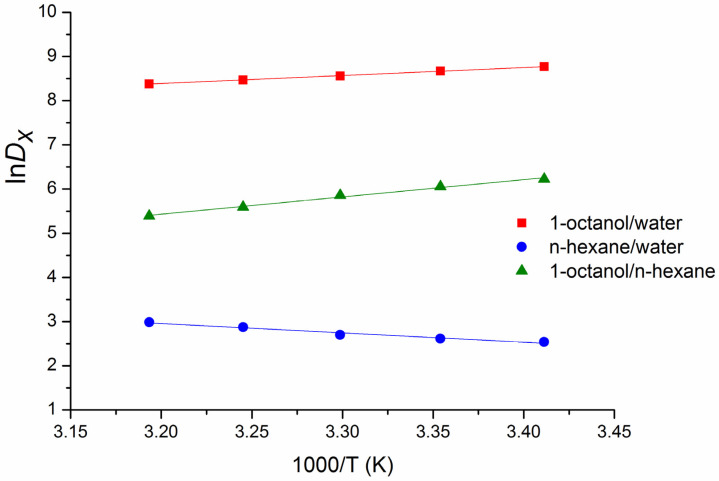
Temperature dependences of the experimental apparent distribution coefficients of BCL in the 1-octanol/water (red color), n-hexane/water (blue color), and 1-octanol/n-hexane (green color) systems (mole fraction scale).

**Figure 5 pharmaceutics-14-00674-f005:**
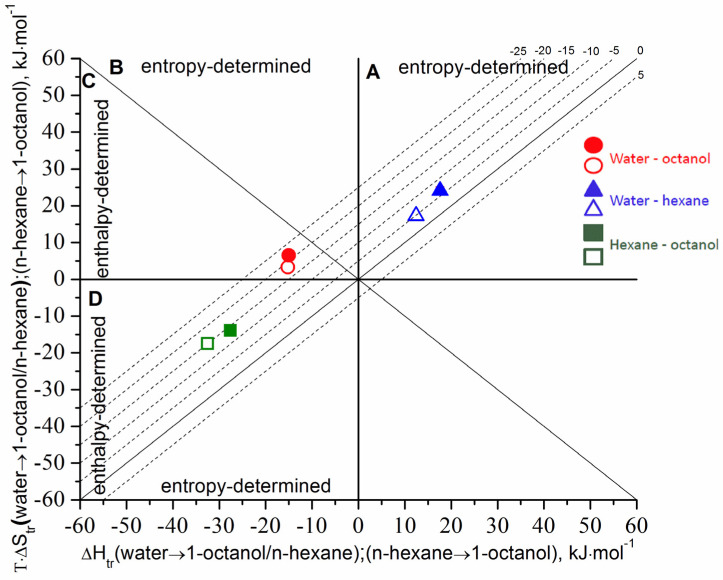
Enthalpy/entropy relationship for the transfer of BCL from water to 1-octanol (red circles) and n-hexane (blue triangles), and from n-hexane to 1-octanol (green squares): hypothetical transfer—the filled symbols; “real” transfer—the empty symbols. The isoenergetic curves of the function are marked by dotted lines.

**Table 1 pharmaceutics-14-00674-t001:** Temperature dependences of solubility (*X*_2_ (mole fract.) and *S*_2_ (mol∙L^−1^)) of BCL in water, 1-octanol, n-hexane, and ethanol at pressure *p* = 100 kPa ^a^.

T (K)	Water	1-Octanol	n-Hexane	Ethanol
	*X*_2_ × 10^7^ (*S*_2_ × 10^6^)	*X*_2_ × 10^4^ (*S*_2_ × 10^3^)	*X*_2_ × 10^6^ (*S*_2_ × 10^6^)	*X*_2_ × 10^3^ (*S*_2_ × 10^2^)
293.15	1.12 (6.24)	2.23 (1.42)	0.74 (5.67)	1.15 (1.95)
298.15	1.47 (8.14)	2.57 (1.62)	1.07 (8.12)	1.42 (2.40)
303.15	1.90 (10.52)	3.00 (1.88)	1.50 (11.33)	1.74 (2.91)
308.15	2.35 (12.95)	3.40 (2.13)	2.04 (15.31)	2.11 (3.49)
313.15	2.97 (16.39)	3.95 (2.46)	2.76 (20.52)	2.43 (3.99)
318.15			3.63 (26.80)	

^a^ The standard uncertainties are *u*(*T*) = 0.15 K, *u*(*p*) = 3 kPa. The relative standard uncertainties are *u_r_*(*S*) = 0.04.

**Table 2 pharmaceutics-14-00674-t002:** Molar volumes and Hansen solubility parameters of BCL and solvents.

Sample	*V* (cm^3^∙mol^−1^)	^a^ $\delta_{d}$ (MPa^0.5^)	^b^ $\delta_{p}\;$ (MPa^0.5^)	^c^ $\delta_{h}$ (MPa^0.5^)	^d^ $\delta_{t}$ (MPa^0.5^)	^e^ $\Delta \delta_{t}\;$	^f^ $\delta_{\upsilon }$ (MPa^0.5^)	^g^*R_a_* (MPa^0.5^)	^i^ RED
BCL	339.7	15.8	13.5	9.0	22.6		20.8	^h^*R_0_* = 11.78	
water	18.0	15.5	16.0	42.3	47.8	25.2	22.3	33.40	2.84
1-octanol	157.7	17.0	3.3	11.9	21.0	1.6	17.3	10.87	0.92
n-hexane	131.6	14.9	0.0	0.0	14.9	7.7	14.9	16.32	1.39
ethanol	58.5	15.8	8.8	19.4	26.5	3.9	18.1	11.41	0.97

^a^ $\delta_{d}=\sum F_{di}/\sum V_{i}$; ^b^ $\delta_{p}=(\sum F_{pi}^{2}{)}^{0.5}/\sum V_{i}$; ^c^ $\delta_{h}=(\sum F_{hi}/\sum V_{i}{)}^{0.5}$; ^d^$\delta_{t}={\left(\delta_{d}^{2}+\delta_{p}^{2}+\delta_{h}^{2}\right)}^{0.5}$; ^e^ $\Delta \delta_{t}=\left|\delta_{t2}-\delta_{t1}\right|$; ^f^ $\delta_{\upsilon }={\left(\delta_{d}^{2}+\delta_{p}^{2}\right)}^{1/2}$; ^g^ calculated by Equation (18); ^h^ calculated by Equation (19); ^i^ calculated by Equation (17).

**Table 3 pharmaceutics-14-00674-t003:** Thermodynamic functions of BCL solubility and solvation processes in the studied solvents at 298.15 K.

Solvent	$X_{2}^{298.15}$	$\Delta G_{sol}^{0}$ (kJ∙mol^−1^)	$\Delta H_{sol}^{0}$ (kJ∙mol^−1^)	$T\Delta S_{sol}^{0}$ (kJ∙mol^−1^)	$\Delta S_{sol}^{0}$ (J∙mol^−1^·K^−1^)	$-\Delta G_{solv}^{0}$ (kJ∙mol^−1^)	$-\Delta H_{solv}^{0}$ (kJ∙mol^−1^)	$-T\Delta S_{solv}^{0}$ (kJ∙mol^−1^)	$-\Delta S_{solv}^{0}$ (J∙mol^−1^·K^−1^)	$\zeta_{H}{}_{solv}^{}\;{}^{a}$(%)	$\zeta_{TS}{}_{solv}^{}{\;}^{b}$(%)
Water	1.47 × 10^−7^	39.0	36.8 ± 0.6	−2.2	−7.4 ± 0.3	24.7	87.9	63.2	212.0	58.2	41.8
1-Octanol	2.51 × 10^−4^	20.5	21.6 ± 0.4	1.1	3.7 ± 3.6	43.2	103.1	59.9	200.9	63.3	36.7
n-Hexane	1.07 × 10^−6^	34.2	49.2 ± 0.7	15.0	50.3 ± 3.4	29.5	75.5	46.0	154.3	62.1	37.9
Ethanol	1.42 × 10^−3^	16.2	28.9 ± 0.9	12.7	42.6 ± 5.1	47.5	95.8	48.3	162.0	66.5	33.5

^a^ $\zeta_{H}{}_{solv}^{}$ = (|$\Delta H_{solv}^{0}$|/(|$\Delta H_{solv}^{0}$| + |$T\Delta S_{solv}^{0}$|))·100%; ^b^ $\zeta_{TS}{}_{solv}^{}$ = (|$T\Delta S_{solv}^{0}$|/(|$\Delta H_{solv}^{0}$| + |$T\Delta S_{solv}^{0}$|))·100%.

**Table 4 pharmaceutics-14-00674-t004:** Transfer thermodynamic functions of BCL at 298.15 K.

Δ*G_tr_* (kJ⋅mol^−1^)	Δ*H_tr_* (kJ⋅mol^−1^)	*T*Δ*S_tr_* (kJ⋅mol^−1^)	Δ*S_tr_* (J⋅mol^−1^⋅K^−1^)	^a^ $\zeta_{Htr}(\%)$	^b^ $\zeta_{TS}{}_{tr}^{}(\%)$	^c^ $\varepsilon_{H}(\%)$	^d^ $\varepsilon_{S}(\%)$
n-hexane → 1-octanol							
−13.7	−27.6	−13.9	−46.6	66.5	33.5	36.6	30.2
n-hexane → water							
4.8	−12.4	−17.2	−57.7	41.9	58.1	16.4	37.4
n-hexane → ethanol							
−18	−20.3	−2.3	−7.7	89.8	10.2	68.8	5.0
water → 1-octanol							
−18.5	−15.2	3.3	11.1	82.2	17.8	-	-

^a^ $\zeta_{Htr}$ =(|$\Delta H_{tr}^{0}$|/(|$\Delta H_{tr}^{0}$| + |$T\Delta S_{tr}^{0}$|))·100%; ^b^ $\zeta_{TS}{}_{tr}^{}$ = (|$T\Delta S_{tr}^{0}$|/(|$\Delta H_{tr}^{0}$| + |$T\Delta S_{tr}^{0}$|))·100%; ^c^ $\varepsilon_{H}$ and ^d^ $\varepsilon_{S}$ were calculated by Equations (20) and (21), respectively.

**Table 5 pharmaceutics-14-00674-t005:** Experimental ($D_{C}^{oct/w}$, $D_{C}^{hex/w}$) and calculated ($D_{C/calc}^{oct/w}$, $D_{C/calc}^{hex/w}$) apparent distribution coefficients of BCL in the 1-octanol/water and n-hexane/water systems at 298.15 K and pressure. *p* = 100 kPa ^a^.

1-Octanol/Water System		n-Hexane/Water System		
^b^ Experimental Distribution Coefficients				
$D_{C}^{oct/w}$	$\log D_{C}^{oct/w}$	$D_{C}^{hex/w}$	$\log D_{C}^{hex/w}$	∆log*D*
662.29 ± 20.22	2.82 ± 0.09	1.88 ± 0.04	0.27 ± 0.01	2.55
^c^ Calculated distribution coefficients				
$D_{C/calc}^{oct/w}$	log$D_{C/calc}^{oct/w}$	$D_{C/calc}^{hex/w}$	log$D_{C/calc}^{hex/w}$	∆log*D_calc_*
199.02 ± 12.00	2.30 ± 0.14	1.00	0	2.30

^a^ The standard uncertainties are *u*(*m*) = 0.01 mg, *u*(*T*) = 0.15 K, and *u*(*p*) = 3 kPa; ^b^ Determined experimentally; ^c^ Calculated from the results of the molar solubility in 1-octanol, n-hexane, and water.

**Table 6 pharmaceutics-14-00674-t006:** Experimental distribution coefficients, ∆ln*D_X_* parameter, and transfer thermodynamic functions of BCL in the 1-octanol/water, n-hexane/water, and 1-octanol/n-hexane systems at 298.15 K and pressure *p* = 100 kPa.

$D_{X}^{org/w}$	$\Delta G_{tr}^{0}$(kJ∙mol^−1^)	$\Delta H_{tr}^{0}$(kJ∙mol^−1^)	$T\Delta S_{tr}^{0}\;$(kJ∙mol^−1^)	$\Delta S_{tr}^{0}\;$(J⋅mol^−1^⋅K^−1^)
^a^ 1-octanol/water system (water → 1-octanol)				
5814.20	−21.5 ± 0.4	−15.0 ± 0.2	6.5	21.8 ± 0.7
^b^ n-hexane/water system (water → n-hexane)				
13.67	−6.5 ± 0.2	17.6 ± 1.8	24.1	80.8 ± 9.9
^c^ Δ$\ln D_{X}^{}$ parameter * (n-hexane → 1-octanol)				
425.33	−15.0 ± 0.4	−32.5 ± 1.6	−17.5	−58.7 ± 4.1

^a^ ln$D_{X}^{oct/w}$ = (2.6 ± 0.1) + (1800 ± 28)/T; r = 0.9997; σ = 6.79 × 10^−5^; *n* = 5 ^b^ ln$D_{X}^{hex/w}$ = (9.7 ± 0.7) − (2115 ± 216)/T; r = 0.9847; σ = 4.15 × 10^−3^; *n* = 5. ^c^ ∆ln$D_{X}^{}$ = (−7.1 ± 0.7) + (3906 ± 197)/T; r = 0.9962; σ = 3.46 × 10^−3^; *n* = 5. * Determined as: ∆ln$D_{X}^{}$ = ln$D_{X}^{oct/w}$− ln$D_{X}^{hex/w}$.

## Data Availability

The results obtained for all experiments performed are shown in the manuscript and SI; the raw data will be provided upon request.

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
