# Peer review of "Physicochemical Profile of Antiandrogen Drug Bicalutamide: Solubility, Distribution, Permeability"

_pharmaceutics, 2022, doi:10.3390/pharmaceutics14030674_

Round 1

Reviewer 1 Report

This is an important study, which highlights pertinent physicochemical properties of the API: bicalutamide (form I). The thermodynamic parameters of dissolution, solvation and transfer process are central data for a deep understanding of solute-solvation interactions.

The article is well structured and easy to read. The conclusions are consistent with the experimental results obtained. Some of the authors have previous experience in this type of thermodynamic approach. In this context, I recommend the publication of the manuscript, after some minor comments that may improve the final text.

Minor comments

Pag. 6, after eq.(2): it should be assumed that the variation of the heat capacity of the solutions with temperature to be negligible. if this is the case, it should be specified in the text.

Pag. 6, eq.(3): please check the equation if a “-“ signal is missing. Also eq. 13 on page 8.

Pag 7 – Introduce a reference relative to Shake-Flask Method.

Pag 9 – Please, introduced some comment on aqueous solubility value of ref [11].

Pag 9 – Why the authors does not give a reference for the use of Hansen solubility parameters in the pharmaceutical area? Just an example: M. Bashimam, Hansen solubility parameters: a quick review in pharmaceutical aspect, J. Chem. Pharm. Res. 7 (8) (2015) 597–599.

Pag 11, fig. 2 – For a better visualization, I suggest remove the full square relative to BIC and introduce some dot/dash lines (vertical and horizontal) for (9.0; 20.8) values.

Pag 20 lines 51 to 54 - Please, check if instead of Figure 4 it should be Figure 5.

Author Response

Reply to Reviewer_1:

This is an important study, which highlights pertinent physicochemical properties of the API: bicalutamide (form I). The thermodynamic parameters of dissolution, solvation and transfer process are central data for a deep understanding of solute-solvation interactions. The article is well structured and easy to read. The conclusions are consistent with the experimental results obtained. Some of the authors have previous experience in this type of thermodynamic approach. In this context, I recommend the publication of the manuscript, after some minor comments that may improve the final text.

Minor comments

Comment:

Pag. 6, after eq.(2): it should be assumed that the variation of the heat capacity of the solutions with temperature to be negligible. if this is the case, it should be specified in the text.

Reply:

The variation of the heat capacity of the solutions with temperature assumed to be negligible. This fact has been specified in the text.

Comment:

Pag. 6, eq.(3): please check the equation if a “-“ signal is missing. Also eq. 13 on page 8.

Reply:

Eqs. 3 and 13 have been corrected.

Comment:

Pag 7 – Introduce a reference relative to Shake-Flask Method.

Reply:

A reference relative to Shake-Flask Method has been introduced in the text.

Comment:

Pag 9 – Please, introduced some comment on aqueous solubility value of ref [11].

Reply:

The comments on BCL aqueous solubility value of ref [11] have been introduced in Results and Discussion.

In addition, another reference including BCL aqueous solubility [Cockshot, I.D. Bicalutamide clinical pharmacokinetics and metabolism. Clin. Pharmacokinet. 2004, 43, 855–878.] has been added.

Comment:

Pag 9 – Why the authors does not give a reference for the use of Hansen solubility parameters in the pharmaceutical area? Just an example: M. Bashimam, Hansen solubility parameters: a quick review in pharmaceutical aspect, J. Chem. Pharm. Res. 7 (8) (2015) 597–599.

Reply:

The additional reference (accompanied by the respective text) for the use of Hansen solubility parameters in the pharmaceutical area has been inserted in the manuscript.

Comment:

Pag 11, fig. 2 – For a better visualization, I suggest remove the full square relative to BIC and introduce some dot/dash lines (vertical and horizontal) for (9.0; 20.8) values.

Reply:

Fig. 2 has been rearranged according to the comment.

Comment:

Pag 20 lines 51 to 54 - Please, check if instead of Figure 4 it should be Figure 5.

Reply:

In order to avoid discrepancies in the discussion of the results, Figure 4 has been checked and reconstructed for clarity.

Reviewer 2 Report

"Review attached"

Author Response

Reply to Reviewer_2:

Comment:

  1. In paragraph 2.1, the words "Solvents and reagents" should be deleted and the purity of n-hexane should be corrected (instead of "0.97%" it should be 97%).

Reply:

In paragraph 2.1, the words "Solvents and reagents" have been deleted and the purity of n-hexane has been corrected.

Comment:

  1. In section 2.2.1, the uncertainty of the temperature of the solubility measurement and the wave numbers used to obtain the calibration curves for BCL in the studied solvents should be given. Moreover, the respective absorption spectra should be presented in the supplementary material.

Reply:

The uncertainty of the temperature of the solubility measurement (±0.1 K) is shown in section 2.2.1.

The absorption spectra in the studied solvents have been presented in the supplementary material (Figure S3).

The absorption spectra are stable, did not reveal any changes upon dilutions.

The wave numbers used to obtain the calibration curves for BCL in the studied solvents were in a good agreement with the literature (Smith, A.A.; K.Kannan, Manavalan, R.; Rajendiran, N. Spectral Characteristics of Bicalutamide Drug in Different Solvents and b-cyclodextrin. J. Incl. Phenom. Macrocycl. Chem. 2007, 58, 161–167.).

The solubility measurements were carried out at a fixed wavelength in each solvent.

Comment:

  1. In equation 6, the symbol ���instead of ��� should be used. Moreover, on page 6,

the font should be corrected and when explaining the meaning of parameter C in eq.7, it would be better to use “the  effect of temperature” instead of “association between the temperatures”.  

Reply:

Equation 6 has been corrected.

When explaining the meaning of parameter C in eq.7, “association between the temperatures” has been replaced by “the effect of temperature”.  

The font on page 6 was a result of converting the manuscript file to pdf formate.

Comment:

  1. In the explanations of the symbols, the authors use the term “apparent distribution coefficient ”. However, the word "apparent" is omitted from the manuscript text. The authors should correct this inaccuracy. In the last sentence on page 7, instead of “the buffer” the word “water” should be written.

Reply:

The term “apparent distribution coefficient ” has been added in the manuscript text.

The word “buffer” has been replaced with the word “water”.

Comment:

  1. On page 8, the sentence “The standard entropy of transfer, , representing the entropy change upon the transferring of one solute molecule was determined by the equation: “ should be corrected. In the sentence “ The samples from the receptor solution (0.5 mL) were withdrawn each 30 min and replaced with an equal amount of the respective fresh buffer.”, instead of “ fresh buffer” should be “fresh water” written.

Reply:

The sentence has been corrected.

The phrase “fresh buffer” has been replaced by “fresh water”.

Comment:

  1. The footnotes to the tables 1, 2 and 5 require some corrections.

Reply:

The footnotes to the tables 1, 2 and 5 have been corrected.

Comment:

  1. In general, the interactions decrease with the temperature growth. Thus, I can not agree with the last sentence in paragraph 3.1.

Reply:

Unsubstantiated information in the last sentence in paragraph 3.1 has been removed.

Comment:

  1. The first sentence in paragraph 3.2 is unnecessarily highlighted.

Reply:

The first sentence in paragraph 3.2 was unnecessarily highlighted as a result of converting to pdf file.

Comment:

  1. In Fig.2 the word “water” instead of “buffer” should be written and in Fig.3, the axis signature should be corrected.

Reply:

In Fig.2 the word “buffer” has been replaced by “water” and in Fig.3, the axis signature has been corrected.

Comment:

  1. In table 4, the units of ��� and ��� are wrong.

Reply:

The units of ��� and  in table 4 have been corrected.

Comment:

  1. On page 14, one of the sentences ”The sublimation thermodynamic parameters determined in our previ-ous study [12] included in (Table S4 Supplementary Materials) were used for the calculation of the solvation parameters (Table 4).” and “At the next step, the solvation thermodynamic functions at 298.15 K (Ta-ble 4) were calculated by Eq. 6 taking into account the sublimation thermo-dynamic functions from our previous study [12] in order to access the inter-actions of the drug with the solvents on the absolute energy scale.” should be deleted as the repetition.

Reply:

The repeated sentence has been deleted.

Comment:

  1. In my opinion the sentences on page 14: “This fact clearly demonstrates the determinative impact of the entropy factor. “ and “Analysis of these values demonstrates he dominant role of the solvation enthalpy…” are mutually exclusive. The authors should comment this fact in the text of the manuscript.

Reply:

The sentences have been reconstructed for clarity.

Comment:

  1. In the footnote of the table 5, the equations defining and are unnecessary as they

are in the text (page 16).

Reply:

The equations defining  and  in the footnote of table 4 have been deleted.

Comment:

  1. The label and the footnote of the table 6 must be corrected.

Reply:

The label and the footnote of table 6 have been corrected.

Comment:

  1. In Fig. 4, the legend is unclear.

Reply:

In order to avoid discrepancies in the discussion of the results, Figure 4 has been checked and reconstructed for clarity including the legend.

Comment:

  1. On page 18, the test must be properly presented. Now does not fit on the page.

Reply:

The text on page 18 does not fit on the page due to unsuccessful converting to pdf file.

Comment:

  1. On page 21, the sentence about the calculation of ∆lnDx must be corrected. The authors say that this property was computed using Equation 11, which is inconsistent with what is written in the footnote of table 7.

Reply:

The sentence about the calculation of ∆lnDx has been corrected according the text written in the footnote of table 6.

Comment:

  1. In SM, the symbols in the S1 table must be corrected, the footnote of the S2 table requires correction, and in the S3 table the number of significant digits in the values of the presented properties and uncertainties should be limited.

Reply:

In SM, the symbols in the S1 table have been corrected, the footnote of the S2 table has been corrected, and in the S3 table the number of significant digits in the values of the presented properties and uncertainties has been limited.

Reviewer 3 Report

Comments                                                                                                      

Volkova et al. submitted the article entitled as “Physicochemical profile of antiandrogen drug bicalu-tamide: Solubility, distribution, permeability” and an attempt has been made to improve solubility and permeability of the drug using various solvents at various temperatures. The manuscript is well designed and informative to reader working related scientific domain. The manuscript needs to be revised before publication. I recommend for major changes and these changes should be carefully addressed in final form.

  1. In section 2.1, Merck source must be included. How did you get “2.1 μS cm-1 electrical conductivity” for bidistilled water. I suggest to write double distilled water. In 2.2.1 authors are directed to mention exact quantity of each solvent taken in each vial. Moreover, there is absorbance wavelength mentioned in the same section.
  2. As per equation 1, it would be better if authors include density of each solvent in table 2. Moreover, Table 2 must be updated with Ra, R and Ra/R values. This would be informative to reader how far solvent is from solute.
  3. In the present manuscript, authors used two deviation parameters such as RAD and RMSD. Both are the same parameters. Why did you calculate both? Clarify. The value of “N” must be mentioned in the manuscript.
  4. In table 1, authors could not study solubility experiment at 313K in water, ethanol and octanol. Why? How did authors validated relative standard uncertainties values?
  5. Please see the sentence “Among the alcohols, a 5.5-fold higher solubility value in ethanol (1.42·10-3 mole fract.) as com-pared to 1-octanol (2.57·10-4 mole fract.)”. What are reasonable factors for improved solubility of the drug in ethanol as compared to water. On comparing the difference of HSP parameters (table 2), there is no significance difference in polarity and hydrogen bonding parameters between the solute and solvent at each temperature. Please justify. Moreover, I suggest to calculate relative Ra/R values for each solvent.  
  6. I suggest to incorporate DSC thermograms of pure and extracted drug from each solvent to identify the stability and polymorphic behavior in explored solvent. This will justify the solid state characteristics in the studied solvent.
  7. The studied solvents such as normal hexane, octanol and ethanol are not solvents for drug delivery formulation. Why did authors select these solvent? I recommend to use GRAS category solvent and biocompatible solvent for administration to patients.
  8. In table 4, authors provided thermodynamic parameters at one temperature points. Why? I suggest to add these functional parameters at other temperature so that readers can understand temperature dependent variations.
  9. Please explain the mechanisms of the drug solubilization in water and ethanol in terms of spontaneous exothermic or endothermic process. Table 4 showed the positive values of Gibbs free energy at explored temperature whereas the same parameter are also positive in ethanol and octanol. Therefore, what was the driving force increased solubility in ethanol and octanol as compared to water.
  10. In table 5, Gibbs energy of transfer coming as negative. How? The result is contradictory from table 4. Figure 4 needs to be rechecked for labeling.
  11. In section 2.2.5, loading drug dose, stirring speed, and lag time calculation. Please mention in the revised form.

Author Response

Reply to Reviewer_3:

  1. Volkova et al. submitted the article entitled as “Physicochemical profile of antiandrogen drug bicalu-tamide: Solubility, distribution, permeability” and an attempt has been made to improve solubility and permeability of the drug using various solvents at various temperatures. The manuscript is well designed and informative to reader working related scientific domain. The manuscript needs to be revised before publication. I recommend for major changes and these changes should be carefully addressed in final form.

Comment:

  1. In section 2.1, Merck source must be included. How did you get “2.1 μS cm-1 electrical conductivity” for bidistilled water. I suggest to write double distilled water. In 2.2.1 authors are directed to mention exact quantity of each solvent taken in each vial. Moreover, there is absorbance wavelength mentioned in the same section.

Reply:

Merck source has been included.

The equipment for electrical conductivity measuring has been indicated in Materials section.

"Bidistilled" water has been replaced by "double distilled water".

Exact quantity of each solvent taken in each vial has been introduced in section 2.2.1.

Absorbance wavelength in each solvent has been transferred from section 3.1. (Results) to section 2.2.1. (Methods).

Comment:

  1. As per equation 1, it would be better if authors include density of each solvent in table 2. Moreover, Table 2 must be updated with Ra, R and Ra/R values. This would be informative to reader how far solvent is from solute.

Reply:

We agree with the reviewer that the densities of the pure solvents should be included. At the same time, the densities were used for the mole fraction calculations at different temperatures. Due to this, it is inconvenient to introduce them to table 2. We tabulated the densities of the solvents at each temperature in a separate table in Supplementary Material file (Table S1).

Ra, R0 and RED=Ra/R0 values have been calculated and introduced in Table 2. The respective explanation has also been provided.

Comment:

  1. In the present manuscript, authors used two deviation parameters such as RAD and RMSD. Both are the same parameters. Why did you calculate both? Clarify. The value of “N” must be mentioned in the manuscript.

Reply:

The relative average deviation (RAD), and root mean square deviation (RMSD) are often used to evaluate the fitting degree of the model. In our study we applied both parameters. Of course, it is possible to use one of them, for ex., RAD.

The value of “N” is mentioned in the Methods section of the manuscript.

Comment:

  1. In table 1, authors could not study solubility experiment at 313K in water, ethanol and octanol. Why? How did authors validated relative standard uncertainties values?

Reply:

Usually, in order to discuss the temperature dependences of the solubility and determine the thermodynamic parameters of the drug compound, the temperature range from 293.15 to 313.15 K is often appropriate. This range includes the standard temperature of 298.15 K and the temperature 308.15 K close to the temperature of healthy humans. The solubility experiments for n-hexane at additional temperature of 318.15 K were carried out exclusively by force of the experimental reasons at that moment.

The standard uncertainty u(S) was estimated as a standard deviation of S. Relative standard uncertainties () were validated from the standard uncertainties () reduced by a mean value of the experimental property () according to the equation:

Comment:

  1. Please see the sentence “Among the alcohols, a 5.5-fold higher solubility value in ethanol (1.42•10-3 mole fract.) as com-pared to 1-octanol (2.57•10-4 mole fract.)”. What are reasonable factors for improved solubility of the drug in ethanol as compared to water. On comparing the difference of HSP parameters (table 2), there is no significance difference in polarity and hydrogen bonding parameters between the solute and solvent at each temperature. Please justify. Moreover, I suggest to calculate relative Ra/R values for each solvent.

Reply:

The factors for improved solubility of the drug in ethanol and 1-octanol as compared to water have been indicated in the text (3.1. Solubility of BCL in Water and Organic Solvents).

Polarity and hydrogen bonding parameters between BCL and solvents are rather different as compared to the parameter of the dispersion interaction.

In order to disclose how far solvent is from solute, relative Ra/R values have been calculated and introduced in Table 2 as RED.

Comment:

  1. I suggest to incorporate DSC thermograms of pure and extracted drug from each solvent to identify the stability and polymorphic behavior in explored solvent. This will justify the solid state characteristics in the studied solvent.

Reply:

The DSC thermograms of pure and extracted drug from each solvent have been added to SI file as Figure S2. In addition, the description of the DSC experiments has been also introduced in Methods section (2.2.4. Differential scanning calorimetry).

Comment:

  1. The studied solvents such as normal hexane, octanol and ethanol are not solvents for drug delivery formulation. Why did authors select these solvent? I recommend to use GRAS category solvent and biocompatible solvent for administration to patients.

Reply:

We agree that the studied solvents are not solvents for drug delivery formulation. Using these solvents is determined by the main object of this study - the investigation of BCL solubility in the media simulating the biological fluids. As follows, we used the appropriate solvents usually applied to simulate the lipophilic medium of the biological cell membranes (1-octanol), non-polar tissues (brain, for example) (n-hexane). Moreover, n-hexane - the inert solvent interacting with the drugs only non-specifically can be successfully applied to disclose the impacts of the specific and non-specific interactions of the drug with the solvents very important part of the thermodynamic investigation. In its turn, ethanol was taken in view of its advantages as a co-solvent, preservative agent and solvent for crystallization.

Comment:

  1. In table 4, authors provided thermodynamic parameters at one temperature points. Why? I suggest to add these functional parameters at other temperature so that readers can understand temperature dependent variations.

Reply:

Thermodynamic parameters at other temperatures have been calculated and presented in Table S6. A short comment has also been introduced in the manuscript.

Comment:

  1. Please explain the mechanisms of the drug solubilization in water and ethanol in terms of spontaneous exothermic or endothermic process. Table 4 showed the positive values of Gibbs free energy at explored temperature whereas the same parameters are also positive in ethanol and octanol. Therefore, what was the driving force increased solubility in ethanol and octanol as compared to water.

Reply:

The explanation of the mechanisms and driving forces of the drug solubilization have been introduced in the text (3.3. Thermodynamics of Solubility, Solvation and Transfer Processes).

Comment:

  1. In table 5, Gibbs energy of transfer coming as negative. How? The result is contradictory from table 4. Figure 4 needs to be rechecked for labeling.

Reply:

In Table 3 the solution free Gibbs energy values (calculated by Eq. 4) are positive for all the studied systems. In their turn, the values of the solvation Gibbs energy calculated by subtracting the respective sublimation parameter from the solution one are obviously negative (the sublimation parameter is greater than the solution one in all the cases). The free Gibbs energy of the transferring from one solvent to another (for ex. n-hexane ® 1-octanol) is calculated by the subtracting of the respective parameter in n-hexane from the same parameter in 1-octanol taking into account the signs ("+" or "-"). In that way, the transferring parameters were derived. We checked carefully the signs of  in Table 4 and got added evidence that the signs are correct. For example,  (n-hexane ® 1-octanol)= (1-octanol) - (n-hexane)=20.5-34.2= -13.7. Similarly, we can calculate the transferring parameters from the respective solvation parameters:

 (n-hexane ® 1-octanol)= (1-octanol) -  (n-hexane)= (-43.2)-(-29.5)= -13.7.

In order to avoid discrepancies in the discussion of the results, Figure 4 has been checked and reconstructed for clarity.

Comment:

  1. In section 2.2.5, loading drug dose, stirring speed, and lag time calculation. Please mention in the revised form.

Reply:

The concentration of the donor solution (M) instead the loading drug dose was used for the Papp calculation and was indicated in the text. The volume of the donor solution was 7 mL. From this the dose can be calculated as 28.86 mg in 7 mL. This dose has been mentioned in the manuscript text.

The stirring speed of 500 rpm has been added in Methods section (2.2.6. In vitro Permeability Experiment).

The lag time was determined to be 30 min since after this moment the permeation rate changed in the range of the experimental error.

Round 2

Reviewer 2 Report

I have read the revised version of the manuscript sent by the authors. I am glad that the authors have improved the article as per my suggestion. MS reads well and seems to me to significantly increase knowledge of the thermodynamics of BCL dissolution and distribution processes in order to assess drug transport properties, partitioning in biological tissues and diffusion across biological membranes. Now, I strongly recommend that you accept your manuscript for publication in Pharmacutics.

Author Response

Thank you very much.

Reviewer 3 Report

Authors revised the manuscript as per suggestion. However, I recommend minor revision for the few sentences and these are as below

Revision: minor revision

The statement “The δt parameter represents the difference between the δt parameters of the solute and the solvent.” should be justified by citing a suitable article such as 1. Afzal Hussain, Obaid Afzal, Abdulmalik S.A. Altamimi, Abuzer Ali, Amena Ali, Fleming Martinez, Mohd Usman Mohd Siddique, William E. Acree Jr, Naushad Ali. Preferential solvation study of (Z)-N-benzyl-2-{5-(4-hydroxybenzylidene)-2,4-dioxothiazolidin-3-yl)acetamide (3) in {NMP (1) + Water (2)} co-solvent mixture and GastroPlus software based in vitro simulation. Journal of Molecular Liquids 349 (2022) 118491. doi.org/10.1016/j.molliq.2022.118491 0167-7322. Similarly, the sentences “From the thermodynamic point of view, the driving force of the dissolution ( DGsol ) 361 is a combination of both the enthalpy and entropy contributions” and “The positive entropy values in the organic solvents impact to favorable dissolution (entropy 367 driven process” should be cited with a reference such as “Afzal Hussain, Mohammad A. Altamimi, Obaid Afzal, Abdulmalik S. A. Altamimi, Abuzer Ali, Amena Ali, Fleming Martinez, Mohd Usman Mohd Siddique,William E. Acree, Jr., and Abolghasem Jouyban ” Preferential Solvation Study of the Synthesized Aldose Reductase 2 Inhibitor (SE415) in the {PEG 400 (1) + Water (2)} Cosolvent Mixture 3 and GastroPlus-Based Prediction ACS Omega 2022, 7, 1, 1197–1210”.

Author Response

Comment:

The statement “The δt parameter represents the difference between the δt parameters of the solute and the solvent.” should be justified by citing a suitable article such as 1. Afzal Hussain, Obaid Afzal, Abdulmalik S.A. Altamimi, Abuzer Ali, Amena Ali, Fleming Martinez, Mohd Usman Mohd Siddique, William E. Acree Jr, Naushad Ali. Preferential solvation study of (Z)-N-benzyl-2-{5-(4-hydroxybenzylidene)-2,4-dioxothiazolidin-3-yl)acetamide (3) in {NMP (1) + Water (2)} co-solvent mixture and GastroPlus software based in vitro simulation. Journal of Molecular Liquids 349 (2022) 118491. doi.org/10.1016/j.molliq.2022.118491 0167-7322. Similarly, the sentences “From the thermodynamic point of view, the driving force of the dissolution ( DGsol ) 361 is a combination of both the enthalpy and entropy contributions” and “The positive entropy values in the organic solvents impact to favorable dissolution (entropy 367 driven process” should be cited with a reference such as “Afzal Hussain, Mohammad A. Altamimi, Obaid Afzal, Abdulmalik S. A. Altamimi, Abuzer Ali, Amena Ali, Fleming Martinez, Mohd Usman Mohd Siddique,William E. Acree, Jr., and Abolghasem Jouyban ” Preferential Solvation Study of the Synthesized Aldose Reductase 2 Inhibitor (SE415) in the {PEG 400 (1) + Water (2)} Cosolvent Mixture 3 and GastroPlus-Based Prediction ACS Omega 2022, 7, 1, 1197–1210”.

Reply:

The statements indicated by the Reviewer have been justified by citing the suitable articles:

Hussain, A.; Afzal, O; Altamimi, A.S.A.; Ali, Ab.; Ali, Am.; Martinez, F.; Siddique, M.U.M.; Acree Jr, W.E.; Ali, N. Preferential solvation study of (Z)-N-benzyl-2-{5-(4-hydroxybenzylidene)-2,4-dioxothiazolidin-3-yl)acetamide (3) in {NMP (1) + Water (2)} co-solvent mixture and GastroPlus software based in vitro simulation. J. Mol. Liq. 2022, 349, 118491, https://doi.org/10.1016/j.molliq.2022.118491 0167-7322.

Hussain, A.; Altamimi, M.A.; Afzal, O; Altamimi, A.S.A.; Ali, Ab.; Ali, Am.; Martinez, F.; Siddique, M.U.M.; Acree Jr, W.E.; Jouyban, A. Preferential solvation study of the synthesized Aldose Reductase 2 inhibitor (SE415) in the {PEG 400 (1) + Water (2)} cosolvent mixture 3 and GastroPlus-based prediction. ACS Omega 2022, 7(1), 1197–1210,

https://doi.org/10.1021/acsomega.1c05788.
